# Neuromodulation for Brain Tumors: Myth or Reality? A Narrative Review

**DOI:** 10.3390/ijms241411738

**Published:** 2023-07-21

**Authors:** Quintino Giorgio D’Alessandris, Grazia Menna, Alessandro Izzo, Manuela D’Ercole, Giuseppe Maria Della Pepa, Liverana Lauretti, Roberto Pallini, Alessandro Olivi, Nicola Montano

**Affiliations:** 1Department of Neuroscience, Università Cattolica del Sacro Cuore, Largo F. Vito 1, 00168 Rome, Italy; quintinogiorgio.dalessandris@policlinicogemelli.it (Q.G.D.); mennagrazia@gmail.com (G.M.); liverana.lauretti@unicatt.it (L.L.); roberto.pallini@unicatt.it (R.P.); alessandro.olivi@policlinicogemelli.it (A.O.); 2Department of Neurosurgery, Fondazione Policlinico Universitario Agostino Gemelli IRCCS, Largo A. Gemelli 8, 00168 Rome, Italy; alessandro.izzo@policlinicogemelli.it (A.I.); manuela.dercole@policlinicogemelli.it (M.D.); giuseppemaria.dellapepa@policlinicogemelli.it (G.M.D.P.)

**Keywords:** brain tumors, glioblastoma, neuromodulation, glutamate receptors, neuroligin-3, deep brain stimulation

## Abstract

In recent years, research on brain cancers has turned towards the study of the interplay between the tumor and its host, the normal brain. Starting from the establishment of a parallelism between neurogenesis and gliomagenesis, the influence of neuronal activity on the development of brain tumors, particularly gliomas, has been partially unveiled. Notably, direct electrochemical synapses between neurons and glioma cells have been identified, paving the way for new approaches for the cure of brain cancers. Since this novel field of study has been defined “cancer neuroscience”, anticancer therapeutic approaches exploiting these discoveries can be referred to as “cancer neuromodulation”. In the present review, we provide an up-to-date description of the novel findings and of the therapeutic neuromodulation perspectives in cancer neuroscience. We focus both on more traditional oncologic approaches, aimed at modulating the major pathways involved in cancer neuroscience through drugs or genetic engineering techniques, and on electric stimulation proposals; the latter is at the cutting-edge of neuro-oncology.

## 1. Introduction

Primary brain and central nervous system cancers represent a substantial source of morbidity and mortality worldwide. The most common and aggressive malignant primary brain tumors are represented by high-grade gliomas: over 12,000 new cases are diagnosed each year. The disease is characterized by systematic recurrence and limited progression-free survival (PFS). The burden of disease is compounded by the fact that, despite the giant strides made in the field, median survival remains 15 months and has not undergone major improvements in recent years [1,2,3,4,5]. Gliomas grow chaotically, destroy the normal brain and generate hemorrhagic and hypoxic/necrotic areas; apparently, such behavior is the opposite of the highly orderly and tightly regulated development of the nervous system. Thus, for a long time, neurogenesis and gliomagenesis were thought to live far apart. At the end of last century, a pool of neural stem cells (NSCs) was discovered in the adult mammalian brain [6]. These cells reside in the subventricular zone of the lateral ventricles and in other niches, such the hippocampus and ventral forebrain, and support a continuous neurogenesis in the adult brain [7]. Intriguingly, about ten years later, a population of tumor-resident cancer stem cells (CSCs) was discovered in malignant gliomas and other central nervous system tumors [8,9,10]. Glioma CSCs are responsible for tumor maintenance, progression and resistance to therapy [11], and are able to phenocopy the tumor of origin when xenotransplanted into immunocompromised rodents [8,9,10]. Notably, CSCs share some key properties with NSCs. These properties include clonogenicity, self-renewal, expression of stemness markers like nestin and CD133 and the ability to differentiate into the range of cytotypes of the normal brain or the tumor, respectively [8,9,10]. These advances first established a clear parallelism between normal neurogenesis and gliomagenesis [12]. An origin of glioma CSCs from NSCs was also hypothesized, with no evidence supporting this assumption.

To describe the irregular glioma growth, it was traditionally postulated that the vectors of tumor spread depended on the availability of vascular support and on the presence of white matter bundles, the latter acting as preferential routes for tumor diffusion [4]. Intriguingly, the improvements in functional MRI imaging have led to the definition of connectomics maps in the human brain, and emerging studies allow one to establish a precise link between such maps and the patterns of growth of gliomas [13]. Thus, accumulating data confirm the profound impact of neurogenesis and neuronal activity on tumorigenesis and the progression of gliomas. As proof-of-concept, a recent study showed that irradiation of the neurogenic subventricular zone ipsilateral to the tumor, while sparing the contralateral one, improved the survival of a cohort of glioblastoma patients [14].

The discovery of the existence of synaptic connections between neurons and glioma cells has opened the way to the research on this novel, exciting topic [15]. The term “cancer neuroscience” has been introduced in the scientific literature to describe this novel field of neuro-oncology [16]. It was demonstrated that long-range neurotransmitter signaling can actively regulate neurogenesis. The main players in this scenario are glutamate receptors, gamma-aminobutyric acid-A (GABA-A) receptors and Neuroligin-3 [16]. Modulating these pathways can profoundly impact tumor growth.

The purpose of the present narrative review was to gather the most recent advances in cancer neuroscience (Table 1). We focused on the main neurobiological features, including organization in communicating networks, that can be seen in the healthy brain and are replicated by the tumor cells of incurable gliomas. In the last part of the review, we described some pioneering electrical neuromodulation techniques with promising effectiveness for brain cancer.

To address this aim, an online literature search was launched on the PubMed/Medline database using the following terms as key words in various combinations: “neuroligin-3”, “AMPA receptor”, “GABA-A receptor”, “neuromodulation”, “brain tumor”, “glioma”, “glioblastoma”. The last search was conducted in April 2023. Two authors (G.M. and Q.G.D.) independently conducted the abstract screening for eligibility. Any discordance was solved by consensus with a third author (N.M). We focused mainly on papers endowed with therapeutic implications and that were published in the last 5 years, since older works have been accounted for in previous reviews. Finally, a narrative review of the most interesting findings was drafted.

## 2. Neurobiological Pathways Underlying Cancer Neuroscience

### 2.1. Glutamate and Glutamate Receptors

Compared with normal astrocytes, glioblastoma cells undergo an expansion of phenotypic properties, including the acquisition of neuron-like signaling [17]. This neuronal signaling includes the upregulation of a specific set of excitable membrane ion channels and synaptic proteins, which has been proposed to enhance the survival and motility of glioblastoma cells. Glutamate, the main excitatory neurotransmitter in the central nervous system, has been found to play a pivotal role in glioblastomas. It promotes proliferation and migration in glioblastoma cells, in addition to its role in normal brain development [17,18]. Glutamate has been found to be highly concentrated in glioblastoma microenvironment, thus fostering intracellular calcium ion accumulation, glioma growth and glioma-related epileptic activity [18].

Pioneering experiments by our group focused on the role of the metabotropic glutamate receptor (mGlur) 3 in maintaining an undifferentiated state in glioma CSCs through the down-regulation of bone morphogenetic proteins and activation of the mitogen-associated protein kinase (MAPK) pathway [19]. Accordingly, the inhibition of mGlur3 in preclinical models led to increased sensitivity to temozolomide, while intratumoral mGlur3 levels were inversely correlated with survival in a glioblastoma patients cohort [20].

In neuroblasts, the activation of mGluRs and N-methyl-D-aspartate (NMDA) receptors stimulates glutamate signaling, resulting in an increase in intracellular calcium levels; this calcium influx plays a crucial role in promoting proliferation and cell survival during neurogenesis [17,18]. Unlike immature neurons, differentiated neurons equipped with α-amino-3-hydroxy-5-methyl-4-isoxazolepropionic acid (AMPA) receptors are impermeable to calcium ions due to the presence of an edited form of the GluR2 subunit. However, neural progenitor cells and oligodendroglial precursor cells express calcium-permeable AMPA receptors that lack the GluR2 subunit or contain the unedited form. Interestingly, glioma cells exhibit high levels of AMPA receptors, but the Glur2 subunit is generally absent; instead, AMPA receptors in malignant gliomas are built up by GluR1 and GluR4 subunits only, and are thus Ca ion-permeable [21]. The increased intracellular calcium influx in glioblastomas caused by the presence of permeable AMPA receptors causes the activation of several key pathways involved in tumor growth, invasion and maintenance, like MAPK and Akt [22]. As a proof-of-principle, the mGlur2 subunit was found at high levels in low-grade gliomas, while it was absent in glioblastoma specimens and glioblastoma-derived cells. mGlur2 overexpression in the U87 glioblastoma cell line inhibited proliferation [23].

Notwithstanding the amount of evidence linking glutamate receptors, intracellular calcium levels and glioma growth and invasiveness, therapies targeting these pathways are currently lacking. The AMPA-receptor inhibitor talampanel [24] and the NMDA-receptor inhibitor memantine were proposed as anti-glioma drugs in early trials, with modest results. Another AMPA-receptor inhibitor, perampanel, has been used in several studies as an add-on drug to treat tumor-related epilepsy, with promising effectiveness [25], and preclinical data show a potential antitumor effect of this drug [26].

### 2.2. GABA-A Receptors

GABA-A receptors rule a key feedback mechanism in the adult neurogenic niche. GABA released by committed neural progenitors activates GABA-A receptors on NSCs, causing cell depolarization with chloride efflux and calcium influx; this leads to a decrease in cell proliferation, crucial for the maintenance of the clonogenic ability of NSCs [52]. The expression of the Na/K/2Cl co-transporter NKCC1, which causes intracellular chloride accumulation, is mandatory for the GABA-A receptor to induce cell depolarization instead of hyperpolarization, thus reducing cell proliferation [16]. This regulatory mechanism is further influenced by modifiers like diazepam-binding inhibitor (DBI), which is abundantly present in neurogenic areas and which inhibits the GABA-A-receptor induced cell depolarization [53]. The depolarizing activity of GABA-A receptors has been reported to decrease cell growth in low-grade gliomas (LGGs) [27]. Conversely, DBI overexpression can drive tumor growth both using GABA-A-dependent mechanisms [16] and using GABA-A-independent mechanisms relying on the modulation of fatty acid metabolism [28].

Recently, the prognostic role of GABA-A has been investigated in two studies focused on low-grade gliomas. Zhang et al. [29] analyzed data from The Cancer Genome Atlas (TCGA) to identify prognostic biomarkers for adult isocitrate dehydrogenase (IDH)-wildtype diffuse low-grade gliomas. Among the GABA-A-receptor subunits investigated, GABRD (one of the nineteen subunits of GABA-A-receptor subunit isoforms) expression was found to be associated with overall survival (OS) in IDH-wildtype low-grade glioma patients. In particular, retained gene expression was associated with longer OS. GABRD expression also showed a negative correlation with tumor-infiltrating macrophages, with the latter usually carrying negative prognostic value, and a specific CpG site, cg13916816, was identified as potentially influencing GABRD expression. More recently, another group independently explored the expression and prognostic value of synapse-associated proteins (SAPs) in low-grade gliomas [30]. Four SAPs, GRIK2, GABRD, GRID2 and ARC, made up a signature correlated with a positive prognostic value for patients affected by low-grade gliomas [29]. However, GABRD was upregulated in low-grade glioma patients with seizures, reinforcing the link already described for glutamate receptors between synapses, gliomagenesis and the pathogenesis of seizures [30]. The therapeutic implications of the presence of GABA receptors in gliomas are currently lacking.

### 2.3. Neuroligin-3

Neuroligins (NLGNs) are post-synaptic adhesion molecules that play a crucial role in synaptic function and plasticity [16]. NLGN1 and NLGN3 are involved in excitatory synapses, while NLGN2 participates in inhibitory synapses. The neuroligins bind to presynaptic partners to perform their functions. While wild-type neuroligins play a central role in normal synaptic function, NLGN3 mutations are linked to altered synaptic function. Interestingly, NLGN3 mutations and amplifications are prominent in pancreatic, prostate and gastric cancers, for which a cancer growth-promoting role of innervation has been widely demonstrated [54].

The pioneering studies by the Monje group published in recent years demonstrated how neural activity can promote the growth and proliferation of malignant gliomas, also exploiting the direct synaptic connections between neurons and glioma cells [15,55].

These studies utilized optogenetic control of neuronal activity in a pediatric glioblastoma xenograft model to show that active neurons promote the proliferation and growth of high-grade gliomas in vivo. The conditioned medium from optogenetically stimulated cortical slices promoted the proliferation of high-grade glioma cultures derived from pediatric and adult patients, suggesting the secretion of activity-regulated mitogens. Among these mitogens, the synaptic protein neuroligin-3 (NLGN3) was identified as the primary candidate, and soluble NLGN3 was shown to be necessary and sufficient for the robust proliferation of high-grade glioma cells. NLGN3 activated the phosphatidylinositol-3-kinase/mammalian Target of Rapamycin (PI3K-mTOR) pathway and induced the feedforward expression of NLGN3 in glioma cells. Notably, higher NLGN3 expression levels in human HGG are associated with poorer overall patient survival. These findings highlighted the role of active neurons in the brain tumor microenvironment. The relationship between neurons and brain cancer is further underscored by perineuronal satellitosis, a hallmark of gliomas characterized by tumor cells clustering around neuronal somata. Though neuronal satellitosis per se is only a pathological finding, it is supposed to mirror the mechanism by which excitatory neuronal activity influences brain cancer growth. This evidence suggests a scenario in which the core physiological function of an organ promotes the growth of a cancer arising within it [55].

In the last 5 years, a large series of papers have been focused on NLGN3 as a key driver of glioma progression (Table 2). Pan et al. demonstrated that the growth of NF1-related optic pathway gliomas relies on NLGN3 [31]. In a genetically engineered mouse model of NF1-associated optic pathway glioma, tumor growth was shown to be dependent on light exposure of the retina of the growing rodent; light deprivation prevented tumor formation and maintenance. In this context, the authors showed that tumor growth was dependent on NLGN3 secretion, and that blocking NLGN3 through genetic knockdown or through ADAM10 inhibition results in the arrest of gliomagenesis.

This evidence has opened the way for different approaches to targeting the NLGN3-dependent altered synaptogenic activity of glioblastomas. The assembly and maintenance of synapses are dynamic processes that require bidirectional contacts between the pre- and post-synaptic structures, and NLGN3 is proteolytically cleaved in response to synaptic activity. There are two cleavage pathways, namely basal- and activity-dependent ones, that produce the mitogenic form of NLGN3. ADAM10, and maybe other proteases, are responsible for basal NLGN3 cleavage, while MMPs are largely responsible for activity-dependent NLGN3 proteolytic cleavage [32]. In vitro, ADAM10 inhibitors can prevent the release of NLGN3. Dang et al. in an elegant study demonstrated the role of the LYN pathway in providing a positive feedback loop able to foster NLGN3 cleavage by ADAM10 [33]. In fact, NLGN3 inhibition by siRNA, besides reducing cell growth and migration, down-regulated epithelial-to-mesenchymal transition pathways and LYN, thus arresting the positive loop. Intriguingly, ADAM10 levels were significantly correlated with overall survival in low-grade, but not high-grade, gliomas [33]. The interplay between NLGN3 and ADAM10 has been exploited in a recent ex vivo organoid study by our group [34]. In that work, we showed that the time required for the integration of glioma CSC spheres into brain organoids depended on the organoids age, being much faster in older than in younger organoids. We postulated that NLGN3 expressed by older organoids was responsible for this behavior and, indeed, the addition of NLGN3 to young organoids fostered CSC tropism for the organoid, while the administration of an ADAM10 inhibitor prevented the integration of glioma cells into mature organoids [34].

An important partner of NLGN3 is represented by Gαi proteins, namely the inhibitory α subunits of G proteins. Specifically, the mediation of NLGN3-induced signaling by Gαi1/3 is crucial for neuronal-driven glioma growth. In both patient-derived and commercial glioblastoma cells, depleting Gαi1/3 effectively inhibits the NLGN3-induced activation of cellular proliferation and migration mechanisms. Conversely, overexpressing Gαi1/3 leads to the opposite [35]. These findings have been validated in rodent orthotopic models. Furthermore, an analysis of the TCGA database revealed the upregulation of Gαi3 in glioma tissues compared with healthy tissues, suggesting a prognostic role for Gαi3 in low-grade gliomas [35]. To sum up, Gαi1 and Gαi3 upregulation is correlated with poor patient survival, high tumor grade and NLGN3 upregulation [35].

Tao et al. explored the interplay between NLGN3 and carbonic anhydrase-related proteins 10 (CA10) and 11 (CA11) [36]. They assumed that, since NLGN3 and other neurotrophins are indispensable for normal brain functioning, they are not optimal candidates to be chosen as therapeutic targets due to the possible unwanted side effects. CA11 is a secreted protein mainly expressed in the brain; moreover, similarly to NLGN3, CA10 and CA11 are ligands of the presynaptic protein neurexin. The authors showed that CA11 secreted in the conditioned culture medium of depolarizing neurons inhibited the proliferation of glioma cells, likely via the Akt signaling pathway. Notably, CA11 in the conditioned medium reduced the production of CA11 by glioma cells, suggesting an autocrine/paracrine regulation feedback. Analysis of tumor tissue and of patients’ survival from independent databases confirmed that CA11 expression is reduced in gliomas and, when expressed, is associated with improved survival [36].

Derks et al. confirmed the link between synaptic activity and brain tumor aggressiveness [37]. Performing a magnetoencephalography in a cohort of 24 diffuse glioma patients candidate for surgery, they showed a significant correlation between low levels of peritumoral and global oscillatory brain activity, low NLGN3 expression and prolonged progression-free survival. Liu et al., in a retrospective cohort of 386 glioblastoma patients, showed that the deep brain regions, which are more likely to host glioblastoma recurrence, display higher levels of NLGN3 than the cortical regions, where tumor recurrence is infrequent [38]. Finally, it has been shown that genetic players involved in synapse formation are required for glial cell proliferation, tumor growth and invasion. In an elegant work using a Drosophila Melanogaster glioblastoma model, Losada-Pérez et al. showed that glioblastoma cells have a signature of post-synaptic cells as compared with presynaptic healthy neurons; however, some glioblastoma cells also showed a presynaptic signature, suggesting the presence of intra-tumoral synapses between glioblastoma cells [39]. Among the post-synaptic genes expressed in glioblastoma cells leading to tumor progression, the NLGN-3 pathways were confirmed to have a key role.

A diagnostic role for NLGN3 was also demonstrated. Serum analysis is a promising tool for the diagnostic and prognostic prediction of glioma because extracellular vesicles carry molecular components from their parental cells. In this context, Wang et al. [40] showed that mRNA levels of NLGN3 in serum extracellular vesicles were significantly higher in glioma patients than in healthy donors, although a notable interpatient variability of NLGN3 levels was noticed.

To conclude, the effect of NLG3, in addition to primitive glial tumors, has also been shown to be critical in neuroblastoma growth [41].

## 3. Electrical Stimulation on Brain Tumors

The advances in the understanding of the electro-synaptic mechanisms underlying glioma progression have revived the interest in electrical-based therapies for gliomas.

The most widely known application of such approaches are Tumor-Treating Fields (TTF), which have shown effectiveness in large trials on newly diagnosed [42] and recurrent glioblastoma [43]. The stimulation paradigm of TTF, however, is quite different from the one provided during standard deep brain stimulation (DBS) for neurological disorders including Parkinson’s Disease. In fact, TTF deliver alternating currents at a higher frequency than DBS (200 kHz) through the scalp of shaved patients. Accordingly, the mechanism of action of TTF is thought to rely on the induction of mitotic arrest and apoptosis of dividing cancerous cells, rather than on the modulation of electrochemical stimuli on glioma cells [42]. From a theoretical viewpoint, the use of an implanted electrode mimicking the DBS setting could provide some advantages compared with TTF. Firstly, very practically, there would not be any more the concern of a bulky hardware to carry and the necessity to shave frequently. Then, biologically, DBS has shown the potential to reduce neuroinflammation [44] and to alter the immunological landscape in an antitumor fashion [45]. In vitro experiments have shown that electric fields given through DBS leads with similar stimulation parameters than DBS, are able to exert antitumor activity. Branter et al. [46] found that electrical stimulation for 7 days at a frequency of 60–190 Hz and at a voltage of 5 V is able to significantly impair the viability and mitotic activity of glioma cell lines. The addition of chemotherapy was able to enhance the antitumor activity of DBS-like stimulation. Moreover, in vitro electrical stimulation of glioma cells, both TTF-like and DBS-like, was able to cause significant changes in gene expression, with down-regulation of players involved in mitochondrial functioning and endoplasmic reticulum stress response. Recently, Hebb and coworkers developed a device for locoregional electrotherapy in brain tumors, which they called Intratumoral Modulation Therapy (IMT) [47,48]. In vitro, monophasic, low amplitude (4 V) and low frequency (130 Hz) electrical pulses given using IMT were able to induce apoptotic death in patient-derived glioblastoma cells [48]. Subsequent elegant in vivo experiments were performed by the same group, implanting F98 rat glioma cells onto the striata of syngeneic rats [47]. For the implant, they used a cannula-bioelectrode construct in which the electrode runs parallel to the cannula used for cell injection: in this manner, the electrode is located at the center of the experimental tumor. Using intermediate frequency currents (200 KHz), the authors demonstrated some reduction in tumor volume compared with non-stimulated tumors, with no side effects. The potential of similar but non-invasive stimulations, such as transcranial electromagnetic stimulation, for the treatment of brain cancer remain to be assessed [49]. Ongoing clinical trials, such as NCT04131862 and NCT04330329, are investigating the effectiveness of electrical neuromodulation techniques, such as transcranial direct current stimulation (tDCS) and deep brain stimulation (DBS), in patients with brain tumors. These studies aim to determine the safety, feasibility and therapeutic potential of electrical neuromodulation in a clinical setting [13,50,51].

Advancements in techniques such as resting-state functional MRI and tractography have allowed researchers to explore the effects of glioma growth on brain connectivity [50,51]. It is noteworthy that gliomas have a far-reaching impact on brain connectivity beyond the physical extent of the tumor itself; the magnitude of this impact has been suggested to be associated with prognostic implications [51]. An integrated analysis has been conducted, examining the relationship between the connectome, brain areas commonly implicated in the development of low-grade and high-grade gliomas and the site-specific gene expression profiles of both normal brain tissue and tumors. This analysis provides a foundation for future investigation, since it is currently impossible to draw definite associations between multimodal markers derived from individuals with different demographic and clinical profiles [13]. In perspective, such studies could have the potential to establish a functional hierarchy of brain regions involved in gliomagenesis with implications for neuromodulation therapies.

## 4. Discussion and Conclusions

According to a systematic analysis from the Global Burden of Disease Study updated in 2016, in recent years central nervous system cancers have gained a larger impact in terms of incidence, deaths and disability-adjusted life years (DALYs). The most common histological type of primary central nervous system cancers are gliomas. The novel World Health Organization classification of central nervous system tumors, fifth edition, has profoundly changed the approach to the definition of the several histotypes, which now relies heavily on molecular features [4]. This is the result of a decades-long approach of dissecting the pathways promoting glioma initiation, growth, invasion and resistance to treatments. Yet, all these advances have not let to an improvement in the prognosis of these tumors, particularly of glioblastoma, which remains dismal. Thus, neuro-oncologists have recently started to look at the brain–tumor interface and, in addition to neuro-immunology, cancer neuroscience has started to emerge. The field of cancer neuroscience is a heterogeneous one. In this review, we have provided a narrative, up-to-date review of the main fields of research in this topic. The first part of the review focuses on a more traditional approach, dissecting the molecular bases of neuron–tumor interplay. Synapsis-related molecules, like glutamate and GABA receptors, and neuroligins, are the main players in this aspect. Thus, a different concept of neuromodulation emerges in which the synaptic activity is modulated by means of pharmacologic or genetic approaches. A more traditional, “electric” neuromodulation glance is at the cutting-edge of cancer neuroscience research, but has promising effectiveness. This field of research warrants further studies to optimize the clinical use of neuromodulation in brain cancer. The updated evidence suggests that it holds significant promise as a future effective therapy.

### Limitations

Since cancer neuroscience is at its beginning, one important limitation of our review is the heavy reliance on preclinical studies conducted using cell cultures and other in vitro models. While these studies have provided valuable insights into the interplay between brain tumors and the normal brain, it is essential to acknowledge that findings from in vitro experiments may not accurately represent the complexity of whole animal biology. Thus, the enthusiasm raised by these novel discoveries is mitigated by the challenges to be faced to translate them into the clinical setting.

## Figures and Tables

**Table 1 ijms-24-11738-t001:** Review layout.

Topic	*N* Reviewed Papers	References
Neurobiological pathways underlying cancer neuroscience		
Glutamate and glutamate receptors	10	[17,18,19,20,21,22,23,24,25,26]
GABA-A receptors	5	[16,27,28,29,30]
Neuroligin-3	11	[31,32,33,34,35,36,37,38,39,40,41]
Electrical stimulation on brain tumors	11	[13,42,43,44,45,46,47,48,49,50,51]

**Table 2 ijms-24-11738-t002:** Studies focusing on NLGN3 as a key driver of glioma progression.

Author, Year	Type of Study	Experimental Design	Results
Pan, 2021 [31]	In vivo (Nf1^flox/mut^; G fap::cre genetically engineered mouse models)	An authenticated mouse model of OPG driven by mutations in the neurofibromatosis 1 tumor-suppressor gene (Nf1) was used to demonstrate that stimulation of optic nerve activity increases optic glioma growth, while decreasing visual experience via light deprivation prevents tumor formation and maintenance.	The formation of Nf1-driven OPGs (Nf1-OPGs) is dependent on visual experiences during the developmental stage, when Nf1-mutant mice are vulnerable to tumorigenesis. If retinal neurons have a germline Nf1 mutation, normal retinal neural activity will cause abnormal NLGN3 shedding within the optic nerve. Blocking the formation and progression of Nf1-OPGs is possible by inhibiting NLGN3 shedding through genetic NLG3 loss or pharmacological methods.
Bemben, 2019 [32]	In vitro (HEK293T or HeLa cells; cultured human or embryonic neurons/wild-type and NLGN3 knock-out mouse brains)	In vitro demonstration that multiple proteases are capable of cleaving NLGN3.	NLGN3 is proteolytically cleaved in response to synaptic activity. There are two cleavage pathways—basal- (ADAM10 is involved) and activity-dependent (MMPs are responsible)—that produce the mitogenic form of NLGN3.
Dang, 2021 [33]	In vitro (U251 and U87 cell lines)	U87 and U251 cell lines were used to (1) assess the expression of NLGN3 in gliomas via IHC (2) to explore NLGN3 function and regulatory mechanisms in those cells with high expressions of NLGN3.	Knockdown of endogenous NLGN3 significantly reduced the proliferation, migration and invasion of glioma cells and down-regulated the activity of the PI3K-AKT, ERK1/2 and LYN signaling pathways. Overexpression of NLGN3 yielded opposite results. LYN functions as a feedback mechanism to promote NLGN3 cleavage. Inhibition of ADAM10 suppressed the proliferation, migration and invasion of glioma cells; opposite this, the expression of ADAM10 was correlated with a higher likelihood of LGG.
Goranci-Buzhala, 2020 [34]	In vitro (co-culture of 20-day, 40-day and 60-day brain organoids with GSC tumorspheres)	Administration of NLGN3 or an ADAM10 inhibitor (GI254023X).	60-day brain organoids are able to incorporate GSC spheres faster than 20-day ones. Administration of NLGN3 to 20-day organoids markedly fosters spheres incorporation. Administration of GI254023X to 60-day organoids markedly slows spheres incorporation.
Wang, 2021 [35]	In vitro (human glioma cells) and in vivo (orthotopic xenograft of GBM and lung cancer cells)	Various genetic strategies were utilized to examine the requirement of Gαi1/3 in NLGN3-driven glioma cell growth.	Gαi1 and Gαi3 play a key role in the signal transduction of several RTKs. Gαi1/3 mediation of NLGN3-induced signaling is essential for neuronal-driven glioma growth. In glioma cells, NLGN3-induced cell growth, proliferation and migration were attenuated by Gαi1/3 depletion with shRNA, but facilitated with Gαi1/3 overexpression. Gαi1/3 silencing inhibited orthotopic growth of patient-derived glioma xenografts and brain metastatic lung cancer in mouse brains, whereas forced Gαi1/3 overexpression in primary glioma xenografts significantly enhanced growth.
Tao, 2019 [36]	In vitro (cell lines and cells derived from GBM patients) and in vivo (mouse models)	CA10 and CA11 expression by cultured neurons within the conditioned medium was assessed; conditioned medium from depolarized neurons was proved to inhibit the growth of glioma cell lines.	Neurons release various positive factors such as NLGN-3 and negative factors such as CA11/CA10 to modulate glioma behaviors. Unknown components in neuronal-conditioned medium inhibit glioma CA11 expression, likely via the AKT signaling pathway. The paracrine CA11 from activated neurons and autocrine CA11 by gliomas coordinate to regulate glioma growth negatively, so that there is a fine-tuned balance between the effects of NLG-3 and CA11. The final balance of these factors in the neuron–glioma microenvironment may determine the readout as oncogenic or tumor-suppressive.
Derks, 2018 [37]	Clinical	24 newly diagnosed patients with diffuse glioma (Grades II-IV WHO CNS 2007 classification) underwent magnetoencephalography (oscillatory brain activity was approximated by calculating the broadband) and subsequent tumor resection. NLGN3 expression in glioma tissue was assessed by IHC.	Lower levels of peritumoral and global oscillatory brain activity were related to lower NLGN3 expression and longer PFS.
Liu, 2018 [38]	Clinical on tumor samples (386 GBM patients) and in vitro (U87 and U251 and patient-derived GBM cell lines)	Cortex neuron culture medium (C-NCM) and basal ganglia neuron culture medium (BG-NCM) were used to cultivate U251, U87 and GBM cells isolated from patients.	In the brain of patients affected by recurrent GBM, NLGN3 levels are higher in deep regions (basal ganglia, thalamus and corpus callosum) compared with the cortex. When the level of NLGN3 was higher, the functional coverage of the cell density of GBM was higher as well (in cultured U87 and U251 cell lines). ADAM10 inhibitors can prevent the release of NLGN3.
Losada-Perez, 2022 [39]	In vivo (Drosophila melanogaster GBM model)	Drosophila GBM model investigated the role of synaptic genes in GBM progression and lethality.	GBM cells have a post-synaptic nature with respect to healthy neurons, and the contribution of post-synaptic genes expressed in GBM cells leading to tumor progression is related, among others, to the NLGN3 pathway. There are intratumoral synapses between GBM cells, and there is a functional contribution of presynaptic genes to GBM calcium-dependent activity and tumor progression.
Wang, 2019 [40]	Serum samples [glioma patients (n = 18) and healthy individuals (n = 9)]	A microbead-assisted method based on flow cytometry was used to estimate the efficacy of EGFR protein expression and *NLGN3* and *PTTG1* mRNA in serum EVs from glioma patients (n = 23) and healthy individuals (n = 12).	mRNA level of NLGN3 was significantly higher in glioma patients than in healthy donors (*p* < 0.01), although the mRNA level of NLGN3 varied between each glioma patient.
Li, 2019 [41]	In vitro (neuroblastoma cells and normal nerve cell HN2) and in vivo (mice)	NLGN3 expression and promotion of neuroblastoma progression was analyzed both in vitro and in vivo.	NLGN3 has a significant role in neuroblastoma growth, activating the PI3K/AKT pathway both in vitro and in vivo.

BDNF, Brain-derived neurotrophic factor; CA11, carbonic anhydrase proteins; CNS, central nervous system; EV, extracellular vesicles; GBM, glioblastoma; GSC, glioblastoma stem-like cells; IHC, immunohistochemistry; LLG, lower-grade glioma; MMP, metalloproteinases; NF1, neurofibromatosis type 1; NLGN3, neuroligin 3; OPG, optic nerve glioma; RTK, receptor tyrosine kinase; PFS, progression-free survival; WHO, World Health Organization.

## Data Availability

Not applicable.

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
