# Peer review of "Neuromodulation for Brain Tumors: Myth or Reality? A Narrative Review"

_ijms, 2023, doi:10.3390/ijms241411738_

Round 1

Reviewer 1 Report

This review describes recent studies related to astrocytomas and more specifically glioblastoma multiforme. It is interesting reading, but there are significant limitations.

The review consists of two halves: the first is a discussion of the association of different proteins markers that are positively or negatively correlated with treatment outcome of brain tumors, and the second part is about the use of electrical stimulation to alter the growth characteristics of brain tumors. The two halves are not joined or related in any way within the review.  As a result, the title of the review is misleading. Neuromodulation applies only to the second part of the review, and none of this is clinically demonstrated, so the question of myth or reality is meaningless - electrical stimulation is a research reality of unknown treatment significance.  But there is no relevant myth to discuss either in terms of neuromodulation or biomarkers of cancer behavior.

More pressing than the title, there is no connection between the discussion of protein biomarkers and electrical stimulation. Is there any evidence that mGlur3 or gaba receptors or neuroligin-3 are related to the response to electrical stimulation? If not, it is hard to see a justification for discussing them together in a single review. The review is fundamentally two different and apparently unrelated topics.

Most of the work is derived from cell cultures and other in vitro studies. There are decades of work in which such studies have been misleading or at best, not representative of whole animal biology. A 'limitations of the methods section' might be in order.

The paper is filled with non-standard abbreviations that make the paper hard to read.  Most of the abbreviations are defined on first use, but some are not. Best would be to spell out more of the abbreviations - why make the reader struggle? Better would be to define them consistently on first use.

In multiple places the authors point out the expression of a protein or an intervention changes the response of the target cells, but the don't say what the change is. this is incredibly frustrating to readers, who have to go look up the paper being referred to in order to understand the meaning the authors are trying (unsuccessfully) to impart. For example, L152. 'GABRD (not defined) were confirmed to carry a positive prognostic role for OS in LGG patients.' Does that mean they positively predict a bad outcome in these patients or does if mean the positively predict a good outcome?  Say what the actual finding was.  Moreover, OS in LGG? Not helpful, these are not standard defintions and they are easily spelled out. You are not saving any trees since these will be read as pdf files, and you are not helping the reader.

Similarly, L173-177. is perineuronal satellitois a good or a bad finding? What is the association actually saying. Also, these results, as described, do NOT demonstrate that excitatory neuronal activity can influence brain cancer. What is described is an anatomical association - not a functional association. If there is a functional association, please tell the reader so that they can track your logic. As written, the statement about satellitosis says nothing about neuronal activity.

Adverbs meant to tweak the readers interest generally fall flat. If you have to tell the reader that something is 'interestingly or intriguingly' worth noting, it probably isn't that interesting or intriguing, or it would be obvious to the reader without your comment. Drop these introductory phrases and let the data speak for themselves, and then the reader can decide what is interesting and intriguing - they don't need or appreciate your guidance.

L215 prognostic role of C alpha i3 - good or bad? Tell the reader what it predicts; do not make the reader look it up for him- or herself.

L227-229. Nothing in what is written establishes a 'fine-tuned balance' between neuroligin and carbonic anhydrase 11; as written they are two independent phenomena. Provide a better description if you want to draw this conclusion.

L300-302. Is DBS-related reduction in neuroinflammation a good thing or a bad thing?  How does reduced neuroinflammation fit with increased T-cell cytotoxicity - aren't these at opposite ends of the neuroinflammatory spectrum?

L331-333. Same issue, what is the impact of gliomas on brain connectivity, and is it good or bad? Just saying there is a relationship is completely uninformative.

The foregoing sections do not support the conclusions or title - the review presents two disjointed and likely unrelated phenomena. If they wish to retain the conclusion, then the connections among the data reviewed must be explained better to establish that some mechanistic connection exists.

Much of the English is non-standard and has not been completely proofread. Some further copy editing would improve the manuscript.

Reviewer 2 Report

1. Line 34 - Is it 12 or 12,000?

2. Heading '2.Results' is misleading as it is a review article. 

3. Font of Table 1 is different from main sections.

4. Conclusion part is missing.

5. Materials and Methods - Flow chart/schematic is greatly appreciated.

6. Discussion part should be expander further.

7. The article's title 'Neuromodulation for brain tumors' - Less focus on neuromodulation.

NA

Reviewer 3 Report

The review article titled "Neuromodulation for Brain Tumors: Myth or Reality? A Narrative Review" and find it to be a valuable contribution to the evolving field of cancer neuroscience. The authors have provided an up-to-date overview of the interplay between brain tumors, with a focus on the malingant gliomas, and the normal brain.

The article highlights the intriguing discovery of direct electrochemical synapses between neurons and glioma cells, which has opened up new avenues for potential therapeutic approaches in the treatment of brain cancers. This novel field of study, termed "cancer neuroscience," offers promising opportunities for advancing our understanding of the underlying mechanisms and exploring innovative treatment strategies.

One strength of this review is its comprehensive coverage of both traditional oncologic approaches and emerging electric stimulation proposals in the context of cancer neuromodulation. By discussing therapeutic avenues ranging from modulating major pathways through drugs or genetic engineering techniques to cutting-edge electric stimulation techniques, the authors provide a balanced and forward-thinking perspective on potential interventions for brain tumors.

The manuscript is comprehesively written abd easily followed. I would recommend that the structrue should be changed to avoid that of a research article. The section results should be changed accordingly and the section materials and methods should be removed and the content could be added in the end of the introduction section.

Overall, the narrative review successfully presents the current state-of-the-art regarding cancer neuroscience and its implications for brain tumor therapy. It effectively conveys the significance of the findings, sheds light on the potential of neuromodulation approaches, and provides a solid foundation for future research endeavors. 

minor language issues that can be easily avoided through a manuscript revision.

Round 2

Reviewer 2 Report

Thanks for modifying the manuscript as per my comments. 

Author Response

Thank you!